

# Generation and characterization of cross neutralizing human monoclonal antibody against 4 serotypes of dengue virus without enhancing activity

Subenya Injampa[1,2], Nataya Muenngern[1], Chonlatip Pipattanaboon[1], Surachet Benjathummarak[1], Khwanchit Boonha[1], Hathairad Hananantachai[2], Waranya Wongwit[2], Pongrama Ramasoota[1,2] and Pannamthip Pitaksajjakul[1,2]

[1] Center of Excellence for Antibody Reserach, Faculty of Tropical Medicine, Mahidol University, Bangkok, Thailand

[2] Department of Social and Environmental Medicine, Faculty of Tropical Medicine, Mahidol University, Bangkok, Thailand

## ABSTRACT

**Background**. Dengue disease is a leading cause of illness and death in the tropics and subtropics. Most severe cases occur among patients secondarily infected with a different dengue virus (DENV) serotype compared with that from the first infection, resulting in antibody-dependent enhancement activity (ADE). Our previous study generated the neutralizing human monoclonal antibody, D23-1B3B9 (B3B9), targeting the first domain II of E protein, which showed strong neutralizing activity (NT) against all four DENV serotypes. However, at sub-neutralizing concentrations, it showed ADE activity *in vitro*.

**Methods**. In this study, we constructed a new expression plasmid using the existing IgG heavy chain plasmid as a template for Fc modification at position N297Q by site-directed mutagenesis. The resulting plasmid was then co-transfected with a light chain plasmid to produce full recombinant IgG (rIgG) in mammalian cells (N297Q-B3B9). This rIgG was characterized for neutralizing and enhancing activity by using different FcγR bearing cells. To produce sufficient quantities of B3B9 rIgG for further characterization, CHO-K1 cells stably secreting N297Q-B3B9 rIgG were then established.

**Results**. The generated N297Q-B3B9 rIgG which targets the conserved N-terminal fusion loop of DENV envelope protein showed the same cross-neutralizing activity to all four DENV serotypes as those of wild type rIgG. In both FcγRI- and RII-bearing THP-1 cells and FcγRII-bearing K562 cells, N297Q-B3B9 rIgG lacked ADE activity against all DENV serotypes at sub-neutralizing concentrations. Fortunately, the N297Q-B3B9 rIgG secreted from stable cells showed the same patterns of NT and ADE activities as those of the N297Q-B3B9 rIgG obtained from transient expression against DENV2. Thus, the CHO-K1 stably expressing N297Q-B3B9 HuMAb can be developed as high producer stable cells and used to produce sufficient amounts of antibody for further characterization as a promising dengue therapeutic candidate.

**Discussion**. Human monoclonal antibody, targeted to fusion loop of envelope domainII (EDII), was generated and showed cross-neutralizing activity to 4 serotypes of DENV, but did not cause any viral enhancement activity *in vitro*. This HuMAb could be further developed as therapeutic candidates.

Corresponding author
Pannamthip Pitaksajjakul,
pannamthip.pit@mahidol.ac.th

## INTRODUCTION

Dengue infection is a leading cause of illness and death in the tropics and subtropics. More than one-third of the world's population lives in areas at risk (*Natasha Evelyn, Mikkel & Annelies, 2013*). DENV is a member of the *Flavivirus* genus in the *Flaviviridae* family. There are four antigenically different serotypes of DENV (DENV-1 to DENV-4) (*Kuhn et al., 2002*). Primary infection with one DENV serotype induces the production of homotypic neutralizing antibodies, which provides lifelong immunity against the infecting serotype. These homotypic neutralizing antibodies can cross-react but less specific against the other three DENV serotypes and persist for a period of several months to a few years (*Guzman & Vazquez, 2010*). Upon secondary infection with heterologous DENV serotypes, these antibodies fail to neutralize virus resulting in an increase of infected cells and higher virus titers. This phenomenon is named antibody-dependent enhancement activity (ADE) (*Halstead & O'Rourke, 1977*). Primary infection typically causes asymptomatic or self-limited dengue fever (DF). Whereas patients in secondary infection usually have an increased risk of developing severe disease, including life-threatening dengue hemorrhagic fever/dengue shock syndrome (DHF/DSS) (*Murphy & Whitehead, 2011*).

Currently, there is no available antiviral drug specific for all four DENV serotypes. Even though there are many studies of neutralizing antibodies against DENV, fully human monoclonal antibodies (HuMAbs) which are able to neutralize all four DENV serotypes without ADE activity are considered to be the main option for passive immune therapy (*Marasco & Sui, 2007*; *Chan & Carter, 2010*; *Low, Ooi & Vasudevan, 2017*).

Previously, we generated HuMAb clone D23-1B3B9 (B3B9) with strong *in vitro* neutralizing activity (NT) against DENV-1 to DENV-4 (*Setthapramote et al., 2012*). However, investigating the viral infection enhancing activity of this HuMAb on Fc-gamma receptor (FcγR)-bearing cells revealed an increase in DENV infection at sub-neutralizing concentrations, which limits its application as a therapeutic candidate (*Sasaki et al., 2013*). To diminish enhancing activity, modifications at glycosylation site Asn297, which influence the binding affinity between Fc region and FcγR presenting on immune effector cells, were established (*Hristodorov, Fischer & Linden, 2013*; *Chan & Carter, 2010*; *Balsitis et al., 2010*).

We constructed plasmids expressing antibody light chain (LC) and modified the Fc region of the heavy chain (HC) constant domain 2 at position N297Q by site-directed mutagenesis. The modified HC plasmid was co-transfected with the LC plasmid into HEK293T mammalian cells to produce full recombinant IgG (rIgG). To evaluate the Fc-modified rIgG as a potential therapeutic candidate for dengue treatment, NT and ADE activity were determined *in vitro* and compared with those of wildtype rIgG. We also demonstrate the mimic binding epitope on envelope protein of dengue virus for understanding specificity and functionality of this HuMAb.

## MATERIALS AND METHODS

### Cell lines and DENV strains

HEK293T cells were maintained in Dulbecco's modified Eagle medium (Gibco, Grand Island, NY, USA) with 10% fetal bovine serum (FBS). For the NT test, Vero cells were cultured in minimal essential medium (MEM) (GE Healthcare UK Ltd., Buckinghamshire, UK) with 10% fetal bovine serum. THP-1 and K562 cells, which were used in ADE assays, were cultured in RPMI 1640 medium (Gibco) with 10% FBS. The DENV strains used in this study were the Mochizuki strain of DENV1, the 16,681 strain of DENV2, the H87 strain of DENV3, and the H241 strain of DENV4. All DENVs were propagated in C6/36 cells, which were maintained in Leibovitz's L-15 medium (Gibco) supplemented with 10% FBS and 0.3% of BACTO Tryptose Phosphate Broth (TPB) (Sigma-Aldrich, St. Louis, MO, USA). CHO-K1 cells were maintained in MEM medium supplemented with 10% FBS and 1% non-essential amino acid (Gibco). All cell lines were kindly provided by Research Institute for Microbial Diseases, Osaka University.

### Generation of wild type and mutated human monoclonal antibody clone B3B9

#### Plasmid construction

Variable HC and LC sequences of B3B9 HuMAb were previously isolated from hybridoma cells and used for construction of HC and LC expression plasmids producing B3B9 rIgG (*Pitaksajjakul et al., 2014*). This HC plasmid was used as a template for N297Q mutagenesis by site-directed mutagenesis with the In-Fusion Cloning System (In-Fusion® HD Cloning Plus; Clontech Laboratories Inc., Shiga, Japan). Primers were designed according to the manufacturer's instructions to mutate the amino acid at position 297 from asparagine to glutamine. This system combines the action of the In-Fusion HD enzyme with inverse polymerase chain reaction (PCR), which generates linearized DNA from a plasmid template. The PCR reaction was composed of the CloneAmp Hifi PCR premix, 300 nM each of forward and reverse primer, 5 ng of plasmid, and distilled water to final volume of 25 µl. The amplification was performed with 35 cycles of 98 °C for 10 s, 55 °C for 15 s, and 72 °C for 5 s. The inverse PCR products were gel-purified using a PureLink® Quick Gel Extraction Kit (Invitrogen, Carlsbad, CA, USA) following the manufacturer's protocol. The linearized plasmids obtained from inverse PCR were then fused by the In-Fusion enzyme, by following the manufacturer's instructions. The reaction was performed at 50 °C for 15 min. The in-fusion reaction was chemically transformed into Stella chemical competent cells. Individual clones were randomly selected for DNA sequencing.

#### DNA sequencing and plasmid preparation

The DNA sequencing of the mutated HC plasmids were confirmed by Sanger sequencing (Macrogen Inc., Seoul, Korea). Potential N-glycosylation site of wild type and mutated B3B9 HuMAb was identified using NetNGlyc 1.0 software (*Gupta, Jung & Brunak, 2004*). Plasmids that contained the target mutation were amplified in *E. coli* and isolated using a

Purelink<sup>TM</sup> plasmid Midiprep kit (Invitrogen) from 100 ml culture for further transfection in mammalian cells.

### Transient expression of N297Q-B3B9 rIgG in HEK293T cells

Plasmids expressing N297Q-B3B9 HC and LC were transfected into HEK293T cells to produce whole rIgG with the N297Q mutation as previously described (*Pitaksajjakul et al., 2014*). The culture medium containing secreted N297Q-B3B9 rIgG was collected and used in immunofluorescence assays (IFAs) to determine the DENV-binding activity. The N297Q-B3B9 rIgG was purified using a protein A affinity column and measured the concentration of N297Q-B3B9 rIgG by BCA protein assay kit (Thermo Scientific, Waltham, MA, USA). The purity of purified antibody was determined by sodium dodecyl sulfate polyacrylamide gel electrophoresis.

### IFA for DENV-binding activity

IFAs were used to determine the binding and specificity of N297Q-B3B9 rIgG for all four DENV serotypes. IFA plates were prepared by infecting Vero cells with DENV at a multiplicity of infection of 0.1 as well as mocked infected. The plates were incubated for 3 days before being fixed and permeabilized with 3.7% formaldehyde and 0.1% Triton X-100, respectively. Culture fluid from transfected cells was added, and the plates were incubated at 37 °C for 1 h. AlexaFluor 488-conjugated anti-human IgG (1:1,000 dilution) (Invitrogen) was added as secondary antibody. The result was observed under fluorescence microscope (IX71, Olympus). For this assay, non-infected cells were used as a negative control.

### HuMAb isotyping

The IgG isotype of the N297Q-B3B9 rIgG was determined by PCR as previously described (*Omokoko et al., 2014*) using complementary DNA isolated from HuMAb clone B3B9 hybridoma as a template. The gene-specific primers used to amplify IgG1, -2, -3, and -4 are listed in Table S1. IgG1 and IgG3 were amplified using the same forward and reverse primers. The primers were used to amplify the hinge region between the Fab and Fc parts that produce different sized PCR products. IgG1 and IgG3 showed specific band at 211 and 346 bp, respectively. With separate PCR reactions, IgG2 and IgG4 were amplified by the same reverse and different forward primers. These IgG2 and IgG4 PCR reactions showed 207 bp and 210 bp, respectively. The PCR reaction was composed of 0.5 $\mu$g of cDNA, 1 mM Tris–HCl (pH 8.0), 5 mM KCl, 1.25 mM deoxyribonucleotide triphosphate (dNTP), 1.25 units of ExTaq DNA polymerase (TAKARA, Shiga, Japan), and 5 mM of each primer, with distilled water to a final volume of 25 $\mu$l. The amplification of each primer pairs was performed in separate reaction with 35 cycles of 94 °C for 30 s, 65 °C for 30 s, and 72 °C for 30 s. The PCR products were separated by agarose gel electrophoresis and visualized by staining with SYBR Safe DNA Gel stain (Invitrogen).

### Foci reduction neutralization test (FRNT)

FRNT assay was performed as previously described (*Pitaksajjakul et al., 2014*). DENV at MOI 0.01 were mixed with different concentrations of purified HuMAbs (B3B9 or

N297Q-B3B9 rIgG) and incubated at 37 °C for 1 h. Each mixture was added to Vero cells in 96-well cell culture plates and incubated for 2 h. The plates were then overlaid with 2% carboxymethyl cellulose (CMC) in MEM medium with 2% FBS and incubated for 2 days for DENV4 and for 3 days for DENV1, 2, and 3. After that, the cells were fixed with 3.7% paraformaldehyde/PBS and 0.1% Triton X-100/PBS. Immunostaining was performed by an incubation with anti-DENV human antibody, followed by Alexa-conjugated anti-human IgG ($H + L$) (1:1,000 dilution). Foci numbers were counted under a fluorescence microscope. The percent reduction was calculated by comparing the foci number for each antibody concentration with the number of foci obtained from a virus–PBS mixture (negative control).

### Antibody dependent enhancement (ADE) assay on K562 cells

The ADE activity of the N297Q-B3B9 rIgG was also assessed on FcγRIIa-bearing K562 cells (*Konishi, Tabuchi & Yamanaka, 2010*). Antibodies at serial four-fold dilutions and viruses were mixed in 10% FBS RPMI medium in 96-well poly-L-lysine-coated plates (Corning Inc., New York, NY, USA) and incubated at 37 °C. After 2 h, 50 µl of $2 \times 10^6$ cells/ml K562 cells were added. The cell–HuMAb–virus mixture was co-cultured at 37 °C under 5% $CO_2$ for 2 days. The cells were fixed with an acetone/methanol fixing solution at $-20$ °C. Immunostaining was performed by adding anti-DENV human antibody and incubating at 4 °C overnight. Horseradish peroxidase-conjugated anti-human IgG ($H + L$) diluted in 0.05% Tween and 1% FBS in PBS was then added, and the samples were incubated at 37 °C for 1 h. The signal was developed with a DAB substrate solution (KPL, Gathersburg, MD, USA). Infected cell counts obtained from the test were adjusted with the mean infected cell counts obtained with the four negative controls set in the same experiment. The cut-off value for differentiating enhancing from non-enhancing activities was calculated from the average plus three times the SD of the percentages of infected cells obtained with the four negative controls.

### Antibody dependent enhancement (ADE) assay on THP-1 cells

DENV at a multiplicity of infection of 0.1 was incubated with serially diluted antibodies in serum-free RPMI medium at 37 °C for 1 h. Then, 150 µl of $5 \times 10^5$ cells/ml THP-1 cells, which express both FcγRI and II on their surfaces, were added and the samples were incubated at 37 °C under 5% $CO_2$. After 2 h, RPMI medium with 4% FBS was added. The cell–HuMAb–DENV solution was incubated for 3 days, after which RNA was extracted from the infected cells using TRIzol® reagent (Invitrogen). The viral RNA was quantified by a one-step Realtime PCR using dengue virus specific primer and glyceraldehyde 3-phosphate dehydrogenase (GADPH) as an internal control (*Sasaki et al., 2013*). The result is presented as the fold enhancement of virus copies compared with the virus–PBS mixture (control).

## Epitope mapping by phage display of random peptide libraries
### Phage affinity selection (Biopanning)

Human monoclonal antibody clone B3B9 that show cross-neutralizing activity to 4 serotypes of DENV was used for mapping the epitope (*Rowley, O'Connor & Wijeyewickrema, 2004*). Panning was performed by using Ph.D.-12 and Ph.D.-C7C Phage

Display Peptide Library Kit (New England Biolabs Inc., Hitchin, UK) according to the manufacturer's instructions. As described in the manufacturer lab manual, the dodeca-and loop-constrained hepta- peptide library consists of $1.2 \times 10^9$ electroporated sequences.

Briefly, 50 µl of protein A/G magnetic beads were blocked with BSA by incubating at room temperature for 1 h, and washed with 0.1% Tris Buffered Saline with Tween (TBST). In the meantime, Phages ($5 \times 10^{10}$ plaque forming units (PFU)) were incubated at room temperature for 30 min with purified B3B9 HuMAb to a final volume of 200 µl TBST. Phage—HuMAb mixture was transferred to the tube containing the blocked magnetic beads and incubated for 20 min at room temperature. After incubation, nonbinding phages that were not captured on magnetic beads were washed with TBST. The bound phages were eluted from magnetic beads with 1 ml glycine elution buffer (0.2 M Glycine-HCl, 1 mg/ml BSA, pH 2.2) and neutralized with 150 µl of 1M Tris–HCl (pH 9.1). Then, the bound phages were directly added to 20 ml early-log Escherichia coli (*E. coli*) ER2738 culture and incubated at 37 °C with vigorous shaking for 4.5 h. After incubation, the culture was transferred to a 50 ml centrifuge tube and spun at 5,000 rpm for 15 min at 4 °C. The phage-containing supernatant was transferred to a new 50 ml centrifuge tube. These amplified phages were precipitated with 20% PEG 8000/2.5 M NaCl at 4 °C overnight, and centrifuged at 12,000 g for 15 min. Phage pellets were then resuspended with 200 µl TBS and used in the next cycle. Three rounds of selection were performed. Specificity of selected phage clones were confirmed by phage ELISA. Briefly, 96-well microtiter plates were coated in triplicate well of 100 µl B3B9 HuMAb at 4 °C overnight. Then, the wells were washed two times with 0.05% Phosphate Buffered Saline with Tween (PBST). After that, the wells were blocked with blocking buffer (3% BSA in TBST). Then, phage lysate of each selected clones from panning step was added to two wells of coated antibody. At this step, wild type M13 phages were added to another HuMAb-coated well as a no fusion peptide control. Consequently, anti-M13 antibody-HRP conjugated; dilute 1:5,000 in blocking buffer were added. All incubations were performed in humidity chamber at 37 °C, for 1 h. After washing, the reaction was developed using $3, 3', 5, 5'$-Tetramethylbenzidine (TMB) substrate (Sigma-Aldrich), and terminated by adding 100 µl of 2.0 M $H_2SO_4$. The absorbance was measured at 450 nm using ELISA reader (TECAN). Plasmids from positive clones of ELISA were isolated and sequenced using -96 gIII sequencing primer 5'-TGA GCG GAT AAC AAT TTC AC-3'. This primer was used to amplify the fusion peptide region. The inserted random 7 and 12 amino acids were identified and analyzed. All inserted sequences were aligned to determine the consensus sequence. The obtained consensus sequences were aligned with dengue viral genome to determine the mimic epitope.

### Phage affinity binding by inhibition ELISA

After sequence analysis, inhibition ELISA was used to confirm candidate phages from each group of inserted peptide sequences that had specifically bound to B3B9 HuMAb. Reciprocal dilution of each phage clones were firstly determined to use in inhibition assay. For inhibition ELISA, B3B9 HuMAb was serially 2-fold diluted started at 20 µg/ml. Then, the diluted antibody was incubated with equal volume of phage at specified dilution at room temperature for 1 h. After incubation, the mixtures were transferred to HuMAb

pre-coated plate, and incubated for another 1 h. Bound phages to the coated HuMAb were detected by anti-M13-HRP, and signal was developed with TMB substrate. The absorbance values were measured at 450 nm using an ELISA reader (TECAN). Absorbance values of each concentration (A) of antibody in solution phase were divided by absorbance values in the absence of antibody (A0), yielding normalized values (A/A0) (*Skottrup et al., 2012*). Non-relevant anti-influenza antibody, which was tested as negative binding with our selected dengue-specific phage clones, was used as a negative control.

### Generation of stable, antibody-secreting CHO-K1 cells

The HC-expressing constructed plasmid (pQCXIP-CH) contained a puromycin-resistant gene and the LC-expressing constructed plasmid (pQCXIH-CL) contained a hygromycin-resistant gene. After determining the optimal concentrations of the corresponding two antibiotics, stable antibody-secreting CHO-K1 cells were selected using puromycin and hygromycin at 8 $\mu$g/ml and 800 $\mu$g/ml, respectively. For stable cell generation, briefly, CHO-K1 cells were seeded in a 6-well cell culture plate the day before transfection to obtain a cell density of 90–95% on the next day. The medium was replaced with Opti-MEM I reduced serum medium. Plasmid DNA was mixed with transfection reagent (Lipofectamine® 2000) and added to the cells, which were then incubated for 5 h. After that, the medium was replaced with culture medium (10% FBS, 1% NEAA MEM). The cells were incubated at 37 °C with 5% $CO_2$ for 24 h. After that, the medium was replaced with culture medium containing the two antibiotics at the concentrations specified above. The medium was changed every 3–4 days until 60% of alive cells were observed. The transfected cells were used for cell cloning by limiting dilution on 96 well-cell culture plates and incubated for 10–14 days with selection media. The positive clones that were able to secrete the target anti-DENV rIgG were selected by IFA using the culture supernatant of each well containing a single stable clone. The cells were scaled up for further characterization. The level of IgG contained in the culture fluid of each positive clone was roughly determined by an IgG quantitation enzyme-linked immunosorbent assay (Bethyl Laboratories, Inc., Montgomery, TX, USA). The functionalities of the purified N297Q rIgG secreted from the stable cells were confirmed by IFAs, NT and ADE assays.

## RESULTS

### Aglycosylated HuMAb clone B3B9 expressed by transient expression system showed cross neutralizing activity to all serotypes of DENV without ADE activity

*Wild type and mutated human monoclonal antibody clone B3B9 were generated and showed cross reactivity to all serotypes of DENV*

The substitution of amino acid at position 297 from Asparagine to glutamine was created and confirmed by DNA sequencing. Five mg of wild type B3B9 and four mg of N297Q-B3B9 HuMAb were obtained. The successful production of wild type and mutant (N297Q) rIgG was verified by the capacity of binding activity observed via IFA. The N297Q-B3B9 rIgG displayed cross-reactivity to all four DENV serotypes similarly to the wild type rIgG (Figs. 1A–1D).

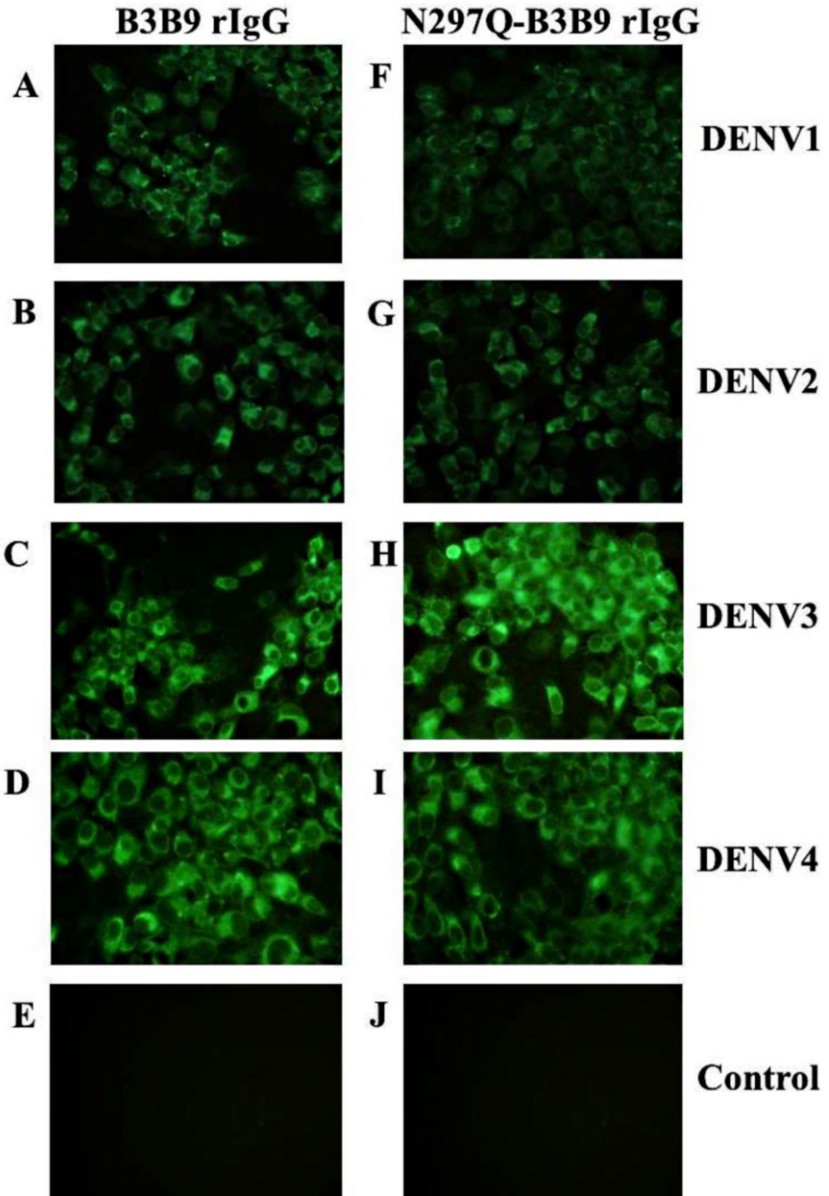

**Figure 1  Immunofluorescence assay of B3B9 and N297Q-B3B9 rIgG against four DENV serotypes.**
Vero cells were infected with DENV1, 2, 3, or 4 at MOI 0.1. The ability of the HuMAbs B3B9 (A–D) and
N297Q-B3B9 rIgG (F–I) in the culture supernatants of HEK293T cells transiently expressing these anti-
bodies to bind to different serotypes of DENV was assessed by performing IFAs. (E, J) Negative control of
mock infected cells. Representative images are shown.

### Human monoclonal antibody clone B3B9 is IgG1 isotype

Reverse transcription PCR was performed to determine the B3B9 rIgG isotype. Using B3B9
hybridoma complementary DNA as template, the PCR product with approximately 211
bp of IgG1 was observed (Fig. S1).
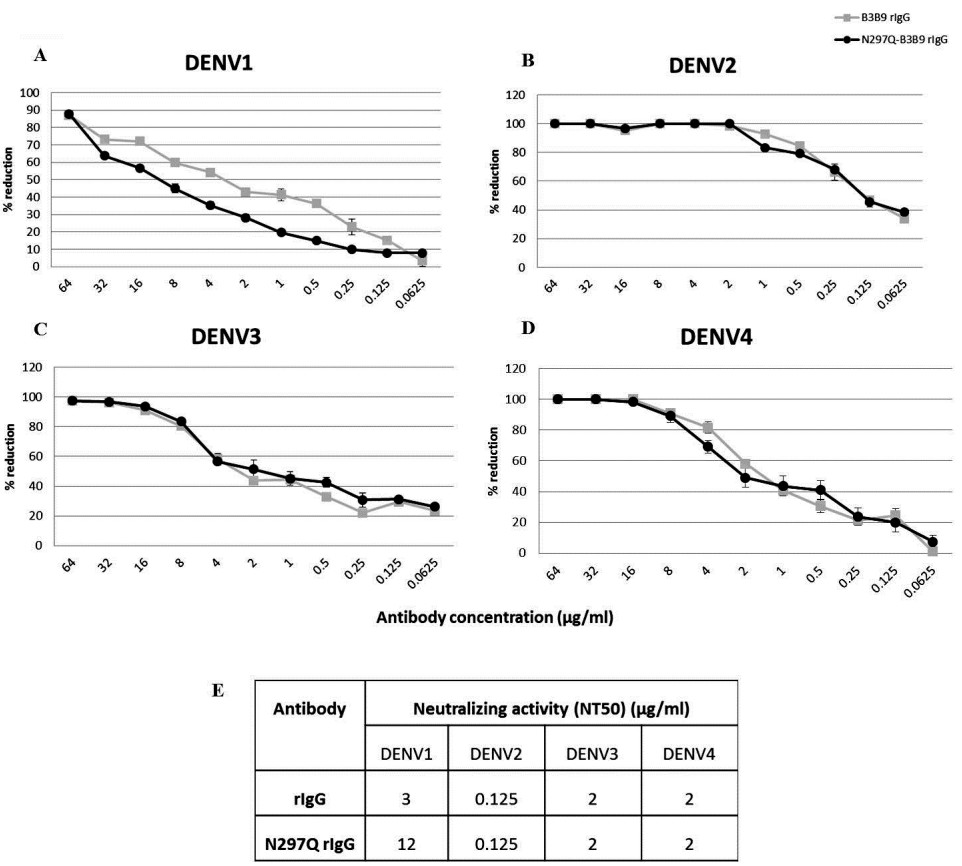

**Figure 2** **Neutralizing activity of N297Q-B3B9 and B3B9 rIgG antibody against four DENV serotypes.** NT levels of N297Q-B3B9 rIgG and B3B9 rIgG of DENV1-4 (A–D) in Vero cells were assessed by foci reduction neutralization tests. The foci of infected cells were counted and compared with the no antibody control, and the results were calculated as the percent reduction in focus forming units. The number of foci was calculated as the average of triplicate experiments. (The error bars show standard deviation of the experiments). (E) NT50 concentration (concentration that was used to reduce half of infected cells when compared with control) of N297Q-B3B9 and B3B9 rIgG antibody against DENV serotypes.

### Wild type and mutated human monoclonal antibody clone B3B9 showed cross neutralizing activity against all serotypes of DENV

To determine the neutralizing activity of rIgG, we assessed the NT of N297Q-B3B9 rIgG and B3B9 rIgG in Vero cells. N297Q-B3B9 rIgG displayed almost the same level of NT as the B3B9 rIgG against all DENV serotypes. The NT levels of various concentrations of these two antibodies against all four DENV serotypes are shown in Figs. 2A–2D. Among these four serotypes, B3B9 and N297Q-B3B9 rIgG showed identical 50% reduction of FFU (NT50) to DENV-2, 3, and 4 (0.125 μg/ml for DENV2 and 2 μg/ml for both DENV3 and DENV4). However, the NT50 concentration against DENV1 of N297Q-B3B9 rIgG was slightly higher than B3B9 rIgG (Fig. 2E).

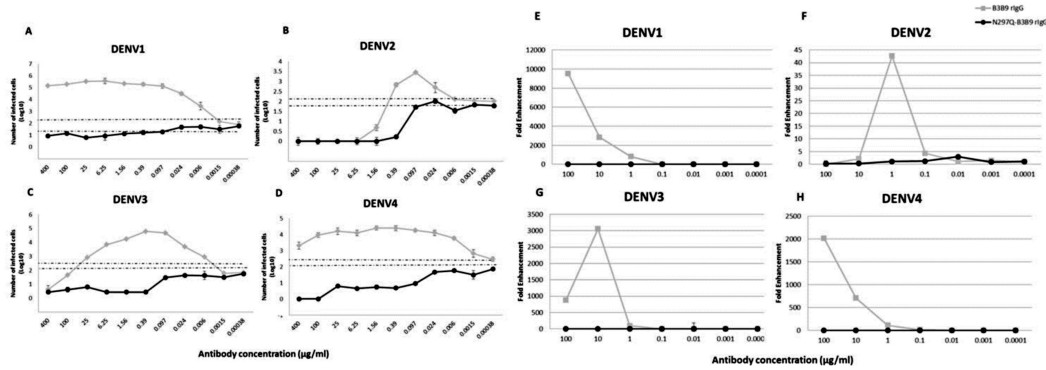

**Figure 3  ADE assays in K562 and THP-1 cells.** (A–D) Enhancement activity against the four DENV serotypes in FcγRII-bearing K562 cells. The number of infected cells ($\log_{10}$) from each antibody concentration compared with that of the control without antibody is shown. The numbers of infected cells were derived from the average of the counted infected cells from three frames at $20\times$ magnification multiplied by a surface area amplification factor to obtain the total number of cells in each well. Dotted lines indicate cut-off values for differentiating neutralizing and enhancing activity from the average plus three times the SD of the percentages of infected cells obtained with the four negative controls. (The error bars show standard deviation of the repeated experiments.) (E–H) ADE assay on THP-1 cells. The antibody concentration was serially diluted ten-fold and are represented on the $X$-axis. The fold enhancement in the virus copy number of the sample with each antibody was compared with that of the no antibody control and is represented on the $Y$-axis. The plotted values were obtained from the average of duplicates from two repeated experiments.

### *Mutated human monoclonal antibody clone B3B9 eliminated ADE activity in K562 cells while wild type showed ADE activity against all types of DENV*

B3B9 rIgG was able to neutralize DENV2 and 3 at the minimum concentration, 0.39 and 25 μg/ml respectively (Figs. 3B–3C), but not neutralize to DENV1 (Figs. 3A) and 4 (Fig. 3D). Moreover, this antibody induced a virus infection enhancement at concentrations 400–0.0015 μg/ml, 0.39–0.006 μg/ml, 100–0.0015 μg/ml, and 400–0.00038 μg/ml for DENV1, 2, 3, and 4, respectively (Figs. 3A–3D). To the contrary, N297Q-B3B9 rIgG showed NT for all four DENV serotypes and showed no enhancing activity in any of the tested antibody concentrations.

### *Mutated human monoclonal antibody clone B3B9 showed no ADE activity in THP-1 cells while wild type showed ADE activity against all types of DENV*

In THP-1 cells, mutant IgG (N297Q-B3B9) that cannot bind to FcγR showed a complete reduction of ADE activity in all tested antibody concentrations. In contrast, wild type rIgG induced 9541, 43, 3061, and 2020 fold enhancement in the dengue virus RNA copy number compared with the control for DENV1, 2, 3, and 4, respectively (Figs. 3E–3H).

## Epitope of B3B9 rIgG was within the highly conserved N-terminal fusion loop peptide of the E protein domain II

Biopanning of a phage display C7C and 12 random peptide library was performed using the affinity selection of purified B3B9 HuMAb. After biopanning, the number of output phages obtained from Ph.D. C7C and Ph.D.12 increased from $6.9 \times 10^5$ to $9.8 \times 10^8$ PFU

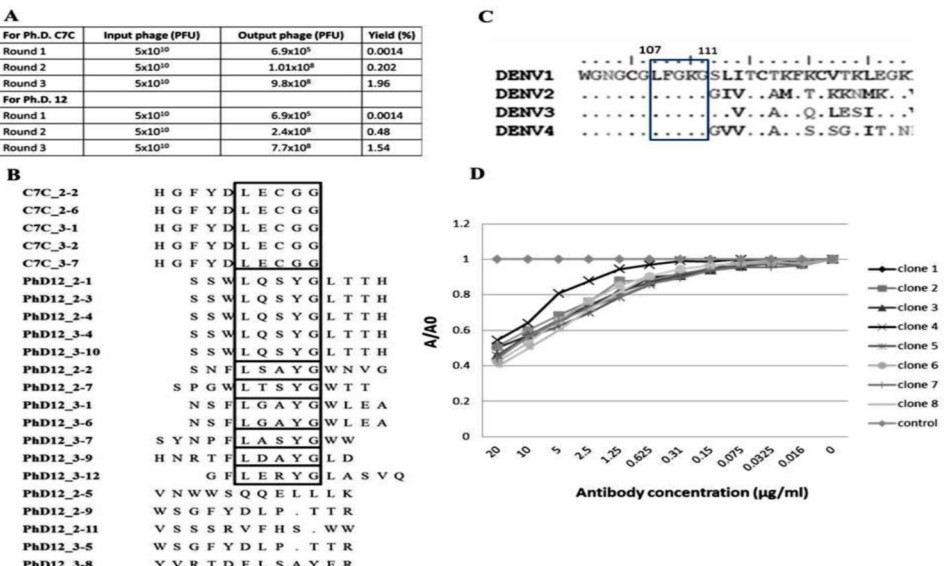

**Figure 4** **Epitope mapping by phage display of random peptide libraries.** (A) Affinity selection of phage-display. Ph.D.-12 and Ph.D.-C7C Phage Display Peptide Libraries were used in biopanning step. The constant units of phage ($5 \times 10^{10}$ PFU) was used for three rounds of biopanning. Increasing percent yields of output phages represents the specific enrichment. (B) Alignment of phage-displayed peptide sequences selected by HuMAbs. Phage clones were shown to display a consensus sequences LXXXG (show in the box). (C) Comparison of the amino acid sequences of E proteins DENV1–4. In the box, DENV1–4 shared the same amino acids at positions 107 (L), 108 (F), 109 (G), 110 (K), and 111 (G). (D) Phage inhibition ELISA of eight phage clones which matched to the motif of 107LXXXG111. Phage lysate of selected clones bound with HuMAb in solution phase and the free phage clones were detected by ELISA. The absorbance values were measured at 450 nm using an ELISA reader (TECAN). Absorbance values of each concentration (A) were divided by absorbance values of no antibody (control) (A0), resulting in normalized values (A/A0). Anti-influenza antibody was used as negative control.

and $6.9 \times 10^5$ to $7.7 \times 10^8$ PFU respectively, which proposed that specific enrichment had occurred (Fig. 4A). Plasmids of phage clones that show binding activity with HuMAb by ELISA were isolated for DNA sequencing. Inserted oligonucleotide sequences of phage DNAs were translated to peptide sequences. Peptide sequences were aligned using BioEdit program 7.2.3 to analyze the epitopes and the binding motif of HuMAb. Phage clones were shown to display a consensus sequence LXXXG (Fig. 4B). After comparing this peptide with the amino acid sequences of E proteins of DENV1–4, it was found that this LXXXG motif matched to 107LFGKG111 within the highly conserved N-terminal fusion loop peptide of the E protein domain II (Fig. 4C). To investigate the solution-phase binding of each candidate phage random peptide with target B3B9 HuMAb, eight candidate phage clones which matched to the motif of 107LXXXG111 were selected for this study (Fig. 4B, Table 1). From phage inhibition ELISA, it was found that the binding of all phage clones were inhibited by B3B9 HuMAb with dose-dependent manner, as shown in Fig. 4D.

**Table 1** **The consensus peptide sequences of eight selected phage clones.** These clones matched to the motif 107LXXXG111 of DENV genome.

| Sequences No. | Consensus sequences |
| --- | --- |
| 1 | LECGG |
| 2 | LQSYG |
| 3 | LSAYG |
| 4 | LTSYG |
| 5 | LGAYG |
| 6 | LASYG |
| 7 | LDAYG |
| 8 | LERYG |

## N297Q-B3B9 rIgG producing from CHO-K1 stable cell lines showed neutralizing activity against DENV2 without enhancing activity

From 125 stable clones that were screened, one showed the highest IgG secretion level (9,587 µg/ml). This clone was continually cultured to collect supernatant for purification. Preliminary study against DENV2 showed that all tested concentrations of the modified rIgG secreted from the stable cell line showed the same result of viral NT (Fig. 5A) without ADE activity (Fig. 5B) as modified rIgG secreted from transient expression.

## DISCUSSION

Due to the complexity of DENV infection and pathogenesis (*Marasco & Sui, 2007*), the ideal therapeutic antibodies should be fully human-derived and capable of inhibiting all four serotypes to reduce the risk of ADE causing more severe symptoms (*Chan, Ong & Ooi, 2013*). We previously generated a cross-neutralizing HuMAb B3B9 (*Setthapramote et al., 2012*). Although this HuMAb showed strong NT to all DENV serotypes, at sub-neutralizing concentrations, it promoted ADE activity (*Sasaki et al., 2013*). To overcome this problem, we generated rIgG with an engineered Fc to prohibit the binding between Fc portion and Fc receptor.

The important receptor for ADE mechanism is FcγR (Fc gamma receptor) that, once it binds with immune complex, can stimulate virus infection. The Fc region of antibody contains asparagine residue (N297) which is a single N-linked glycosylation site in its CH2 domain. The nature of this glycan can decisively influence the therapeutic performance of a recombinant antibody, and their absence or modification can leads to changing of conformation and losing Fc effector functions (*Hristodorov, Fischer & Linden, 2013*; *Subedi & Barb, 2015*). The Fc fragment of antibody acquires the sugar moieties attached at position N297 residues to maintain the structure of antibody. Thus, deletion of the N- glycan changes the structure of the Fc portion resulting in diminished binding to Fcγ receptor (*Nimmerjahn & Ravetch, 2008*).

Firstly, we evaluated the generated N297Q-B3B9 rIgG for its neutralizing activity in Vero cells and found that the engineering antibody showed proper neutralizing activity comparable to B3B9 rIgG against DENV2, -3, and -4 with some lesser degree of NT activity against DENV1 of N297Q-B3B9 comparing with B3B9 rIgG.

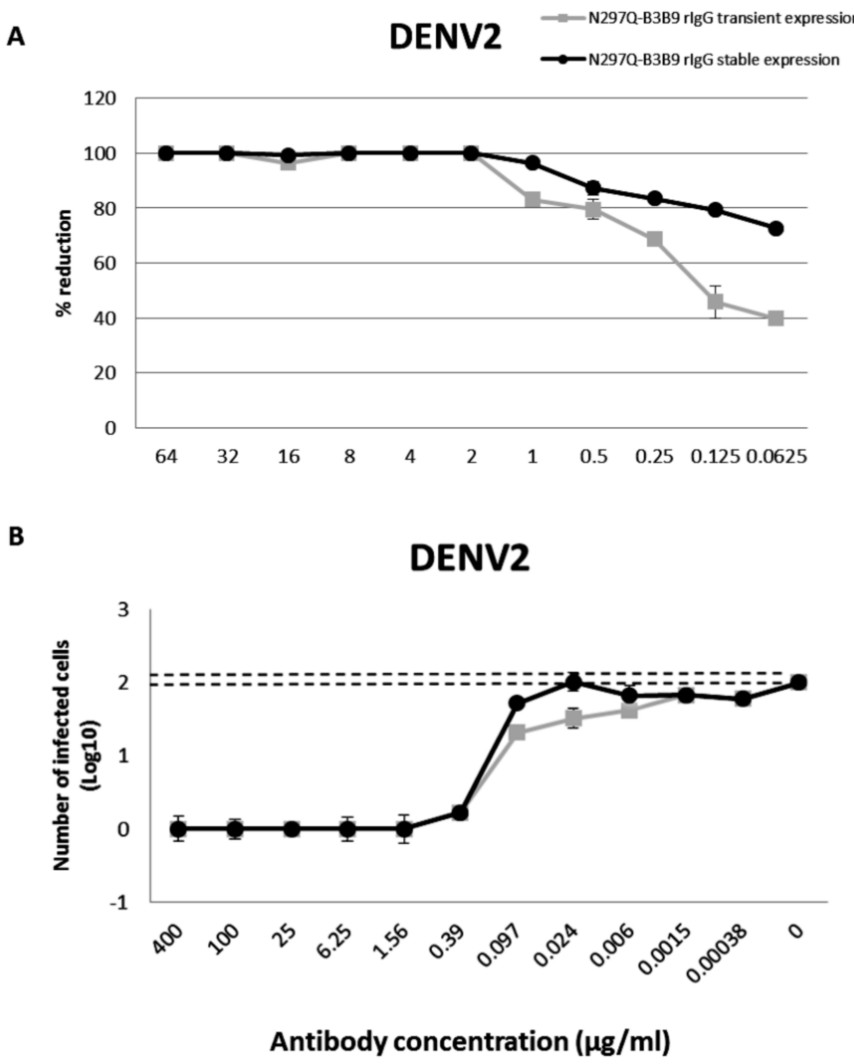

**Figure 5 The NT and ADE activity against DENV2 of N297Q-B3B9 rIgG derived from a stable and transient CHO-K1 cell line.** A line of CHO-K1 cells stably expressing N297Q-B3B9 rIgG was established. (A) The NT of N297Q-B3B9 rIgG from different expression system against DENV2. The $X$-axis represents the concentration of antibodies, and the $Y$-axis represents the percent reduction in focus-forming units compared with the foci number of the control sample that lacked antibody. The number of foci was calculated as the average of triplicate experiments. (B) An ADE assay of N297Q-B3B9 rIgG from different expression system against DENV2 was performed in K562 cells. The infected cells were counted at each antibody concentration and compared with the control (DENV2-infected cells without antibody). Dotted lines indicate cut-off values for differentiating neutralizing and enhancing activity from the average plus three times the SD of the percentages of infected cells obtained with the four negative controls. (The error bars show standard deviation of the repeated experiments.)

Since most of anti-dengue virus antibodies could showed enhancing activity at sub-neutralizing concentration (*Sasaki et al., 2013*; *Schmidt, 2010*). The study in the cell absent of FcγR may not represent the neutralizing status *in vivo*. We then studied the balance of NT and ADE activity using K562 cells, which express FcγRII. In the simplified ADE assay using K562 cells (*Konishi, Tabuchi & Yamanaka, 2010*), the virus infection enhancement is represented as the number of infected cells from each antibody concentration compared with those from a control performed in the absence of antibody. In this cell type, we found that B3B9 rIgG, which showed cross-neutralizing activity to 4 DENV serotypes in Vero cell, only neutralized DENV 2 and 3, but not DENV1 and 4 (Figs. 3A–3D). In contrary, N297Q-B3B9 rIgG showed cross-neutralizing activity to 4 serotypes of DENV.

However, since K562 cell only expresses FcγRIIa, we further determined the ADE activity in THP-1 cells that expressed both FcγRI and FcγRII. In accordance with ADE study in K562 cells, our N297Q-B3B9 rIgG showed clearly reduction of viral enhancement activity in THP-1 cell. This result showed the potential of N297Q-B3B9 rIgG as a human monoclonal therapeutic antibody that cross neutralizes all serotypes of DENV without enhancing activity.

Many studies of Fc modification were explored for establishment of therapeutic antibodies candidates against dengue virus. In 1989, *Tao & Morrison (1989)* studied roles of aglycosylated chimeric mouse-human IgG antibody by modifying at glycan positions N297Q, N297H and N297K via site directed mutagenesis. These aglycosylated antibodies cannot bind to human FcγRI and not trigger C1q binding ability of complement system (*Tao & Morrison, 1989*). *Balsitis et al. (2010)* reported that a N297Q mutation of mouse and chimeric human-mouse IgG that efficiency reducing ADE *in vitro* could decreased the mortality of DENV-infected mice (*Balsitis et al., 2010*).

The results of this study differ to the one carried out by *Ramadhany et al. (2015)*, which modified human monoclonal antibody at N297A and showed NT activity to be the same as its parental HuMAb. Nonetheless, this HuMAb still induced low levels of virus infection enhancement (*Ramadhany et al., 2015*). One possible reason might be the type of mutated amino acid used. Considering the amino acid structure, the substitution of Asparagine with Glutamine (conservative mutation) reduces the effects of functional properties because the side chains of these two amino acids differ by only one methylene group. Thus the substitution amino acid is one of the essential factors that affect to the efficiency of antibody.

Several studies have been approved for successful utilization of random peptide phage display for finding specific epitope to several antiviral (*Xue et al., 2012*; *Zhao et al., 2012*), and anti-flavivirus monoclonal antibody (MAb) (*Sun et al., 2011*). This technique provides an economical and rapid approach for mapping antibody epitopes (*Zhang et al., 2006*; *Chin et al., 2017*). We mapped the epitopes of our HuMAb to LXXXG which correspond to 107LFGKG111 located in the conserved N-terminal fusion loop of envelope domain II (EDII). In accordance with our previous studies, this cross-neutralizing HuMAb B3B9 also targeted to DII of envelope proteins residue 52–132, analyzed by western blot using truncated E protein (*Sasaki et al., 2013*). This is a major target epitope of human antibodies for NT and ADE activity (*Costin et al., 2013*; *Deng et al., 2011*). Comparing epitopes which

targeted murine and human MAb specific to dengue virus (*Shrestha et al., 2010*; *Sukupolvi-Petty et al., 2010*; *Schieffelin et al., 2010*; *Beltramello et al., 2010*; *De Alwis et al., 2012*), it was found that MAbs generated from mice are mostly serotype-specific that targeting DIII of envelope protein (*Shrestha et al., 2010*). However, most of the HuMAbs were targeted to DI-II of envelope proteins which is more cross-reactive (*Beltramello et al., 2010*; *De Alwis et al., 2012*).

As a promising therapeutic candidate, it was described by *Williams et al. (2013)* that the chimeric N297Q MAbs targeting fusion loop and dimer interface on EDII, but not the A strand and C-C' loop on EDIII, acted therapeutically by competing against enhancing antibodies in polyvalent serum that recognize the same or proximal epitopes. This kind of fusion loop specific N297Q MAb showed protective activity *in vivo* when administered with enhancing titer of polyserum (*Williams et al., 2013*). Thus, the identification of B cell epitope in this study is crucial to understand its function and antibody/epitope interaction.

For antibody production, in this study, we used HEK293T cell for transient expression mainly because of its higher transfection efficiency and expression level, resulting in lower development costs when compare to stable cell line development (*Zhang & Shen, 2012*). However, for further characterization of N297Q-B3B9 rIgG as a dengue therapeutic candidate large quantities of antibody, with high stability in both production yield and quality, were required. Therefore, stable expression was used as an alternative solution for antibody production. CHO-K1 cells were used for stable cell line generation. This cell line is the prominent system for bio-manufacturing of therapeutic products (*Hossler, Khattak & Li, 2009*) as 70% of the therapeutic protein was produced by this system (*Croset et al., 2012*). Characterization of this antibody in the other aspects such as stability, pharmacokinetics or phamacodynamics of antibody (*Liu, 2015*) was required because protein produced by HEK293T cells and CHO-K1 cells show different patterns of glycosylation. However, we found no differentiation of NT and ADE activity in both cell types (Fig. 5).

Together, our results suggest that N297Q-B3B9 rIgG is a human monoclonal antibody that can neutralize all four serotypes of DENV without viral enhancing activity. As a fully human-derived monoclonal antibody, it avoids the problem of a human anti-mouse antibody response.

Interaction of Fc-FcγR on innate immune cells trigger immune effector functions such as antibody dependent cell-mediated cytotoxicity (ADCC), complement-dependent cytotoxicity (CDC) (*Kellner et al., 2014*), and antibody dependent cellular phagocytosis (ADCP) (*Grevys et al., 2015*). After thoroughly characterizing these functions, this B3B9 rIgG can be a model testing for dengue therapeutic treatment.

## ACKNOWLEDGEMENTS

The authors would like to thank Dr. Atsushi Yamanaka for his continuous advisement and encouragement.

### Funding

This work was supported by the Faculty of Tropical Medicine, Mahidol University grant number 04/2556 and the National Research Council of Thailand grant number 32/2558. The Deutscher Akademischer Austauschdienst (DAAD, German Academic Exchange Service) fund for the student scholar grant number 577177105. There was no additional external funding received for this study.

### Grant Disclosures

The following grant information was disclosed by the authors:
Faculty of Tropical Medicine, Mahidol University: 04/2556.
National Research Council of Thailand: 32/2558.
The Deutscher Akademischer Austauschdienst: 577177105.

### Competing Interests

The authors declare there are no competing interests.

### Author Contributions

- Subenya Injampa conceived and designed the experiments, performed the experiments, analyzed the data, wrote the paper, prepared figures and/or tables.
- Nataya Muenngern performed the experiments, analyzed the data, wrote the paper, prepared figures and/or tables.
- Chonlatip Pipattanaboon conceived and designed the experiments, performed the experiments.
- Surachet Benjathummarak, Khwanchit Boonha and Hathairad Hananantachai performed the experiments.
- Waranya Wongwit reviewed drafts of the paper.
- Pongrama Ramasoota conceived and designed the experiments, contributed reagents/materials/analysis tools, reviewed drafts of the paper.
- Pannamthip Pitaksajjakul conceived and designed the experiments, performed the experiments, analyzed the data, contributed reagents/materials/analysis tools, wrote the paper, reviewed drafts of the paper.

### Data Availability

The raw data used to generate statistical analysis have been uploaded as Supplemental Files.

### Supplemental Information

Supplemental information for this article can be found online at http://dx.doi.org/10.7717/peerj.4021#supplemental-information.

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
