# Peer review of "Generation and characterization of cross neutralizing human monoclonal antibody against 4 serotypes of dengue virus without enhancing activity"

_PeerJ, doi:10.7717/peerj.4021_

## Round 0.1 · original submission · Major Revisions

I particularly recommend a through revision of spelling and grammar, either by a colleague native speaker, or from a professional service. In its current form, this manuscript does not show proper technical soundness, lacking several controls and experiments. In summary, should you address all suggestions and experiments recommended by both reviewers, I will be happy to receive your revised version.

Reviewer 1 ·

Basic reporting

The report is clear and contains relevant results. However, language could be improved and revision is needed.

Experimental design

The research question is well defined and the results support the conclusion. Some terminology would have to be spelt out.

Validity of the findings

Although the study data is robust, the impact and novelty of the results are not immediately clear. While there are many findings on neutralizing antibodies to DENV for therapeutic usage, the authors would need to highlight the utility and novel functions of the antibody in this study.

Additional comments

Major
(1) The manuscript was well written and the experiments are well defined and support the conclusion of the study. While neutralizing antibodies are critical in disease protection, your introduction needs to be further clarified as to why the epitope characterization is needed, as some part of the study has been done by other investigators. The objective would need more clarity, for example, are you performing the experiments to develop a potent neutralizing antibody with no ADE activity or are you trying to determine the antibody epitopes?
(2) Because the antibody used was developed from a previous study, please clarify which parts of the study (if any) contains data from the prior study.
(3) It is known that ADE activity hampers neutralizing activity of an antibody. Please explain why both the modified and original antibodies demonstrated similar NT levels, despite the absence of ADE activity in the modified antibody.
(4) Would the concentration of antibody needed for virus neutralization/disease protection be feasible? How much antibody would be required to neutralize DENV at different MOI?
Minor
(1) Try not to use abbreviations in the title, it would be better to spell out ADE.
(2) The manuscript needs to be revised for spelling and grammar. Some examples where language could be improved include lines 239, 246, 292, 319.
(3) Describe PhD-12 and C7C. Were they the identification codes for phage display?

·

Basic reporting

Paper is poorly written, English is not clear and unprofessional. Many grammatical, incomplete sentences, poorly chosen words, and conceptual mistakes (e.g humeral instead of humoral) of the English language are found through all the paper’s length. For example (and only 3 examples):
Lines 108-109 Purified B3B9 HuMAb was used in phage display panning experiment to characterize their binding epitopes
Lines 286-288: After comparing this peptide with the amino acid sequences of E proteins of DENV1–4, it was found that all these LXXXG motif was matched to 107LFGKG111…
Lines 393-394: Moreover, N297Q mouse MAb can competed enhancing antibody in polyvalent serum in in vitro study (Williams et al., 2013).

A proofread procedure by an English native speaker is strictly needed.

The title should include the specificity of the MAb characterized to enhance the relevance of the study. For example: “Characterization of aglycosylated human monoclonal antibody targeting the fusion loop of dengue E protein shows strong neutralizing and no enhancing activity“

Introduction:
The introduction is lacking important information:
1. The IgG glycosylation patterns and its interaction with the Fc-gamma receptor is needed in order to expose the importance of the Fc modification in the Mab. Literature from Nimmerjahn F and Ravetch is suggested.
2. “Dengue disease, transmitted by mosquito-borne infection,…” is poorly described. Dengue virus is transmitted by a mosquito, dengue disease is the result of this infection.
3. The aim of the study written in the last paragraph is confusing and a stronger statement regarding the title of the manuscript is suggested. Through all the manuscript the main objective of the paper is not clear. Even though there was a lot of work invested and the results are interesting, the paper lacks the proper communication of its message and there is not a clear relevance of its findings.

Figures:
Figure 1c, 2 are not of high quality, they look blurry.
In Figure 2 is the control just PBS instead of the antibody, a proper control is a staining including non-infected cells.
In Figure 4. Please show first the ADE in K562 and in B the THP-1 as it is explained in the text. It may seems that A is B and B is A if the methods and results portion is correct. K562 ADE was measured by counting infected cells and ADE in THP-1 with RNA quantification. Which by the way, is not a quite accurate method for measuring Dengue, since there are a lot of RNA products that are not part of an infectious virus particle.

Hypothesis and scientific question is not clearly expressed and needs to be more accurate.

Experimental design

Even though the paper is interesting, it is a purely an instrumental paper constructing a potential anti-dengue therapeutic antibody, but it is of concern if it is enough for the PeerJ journal. Maybe if the paper would be improved in clearly stating its objective, it can be taken for consideration for publication.
A more profound justification of the reason for choosing the Phage display methodology is needed. Concentrations and virus moi are missing. In Figure 1, an antibody without the specificity in the competititve Elisa is needed as a negative control.
The research question is not well defined. From the results, the reader can infer the biological relevance of the paper but it is not clearly stated.
The exact reference in some cases is missing, for example, on has to infer the original work showing the origin and description of the Mab B3B9.

Results are written with scarce findings and conclusions. Some controls are lacking. For example:
- demonstrating the absence of glycosylation in the Ab is needed.
- showing the non/binding to the Fc gamma receptors by FACS is missing.
- The specificity of the MAb should be confirmed by an assay showing the interaction between the MAb and the fusion loop.
- Treatment of the original MAb with enzymes that removes the glycosylations would give more robustness to your results.

Validity of the findings

The potential impact and novelty in this manuscript is not assessed by that authors. Even though there is a potential newly produced therapeutic antibody, the paper lacks to send its message.
The asseverations of having constructed a new therapeutic antibody are ambitious regarding the results that they are showing. No proper controls are shown. The biological relevance of an antibody cannot be limited only to its neutralizing or Fcgamma receptor binding. Therefore, the paper needs to be concluded in a more conservative way.

Additional comments

Even though the paper is not well written, the message is not clear and the biological relevance of this new antibody is purely speculative, if the paper is rewritten in a clearly manner, some of the methods and results are more extensively described, if in the discussion they will be able to suggest the potential of this Ab but lowering their asseverations, the paper could be published.

---

## Round 0.2 · Minor Revisions

Experimental details were mostly addressed, however, it is compulsory to have the manuscript revised by a native speaker. Other than this, it is necessary to also respond to concerns provided by the reviewer.

·

Basic reporting

The manuscript was improved, but it is still no publication material. Even though they included necessary information sometimes ill positioned in the manuscript, English is still not at a level for publication. For example, and just one example of many throughout the manuscript, cells do not "contain" a receptor, cells express a receptor. Scientifically the paper is ok, but the written manuscript is poorly explanatory, with many grammatical mistakes, many sentences without any sense.
I still need a more thoroughly revision of the literature about Glycosylation of antibodies and how this affects their function. Also, why the phage displayed was used for the detection of the antibody specificity.
In the results section, the subtitles have to show a finding, not merely described what they have done to achieve this finding.
Negative controls are intended to be part of the figures. In the IFA figure, the negative control of mock infected cells detected by the antibodies are not included.
ADE was not originally described by Schmidt, but by Halstead. Please revise. It is Ab dependent enhancement. please revise through all the MS.
I still think that the paper is publicable, but has to be improved.

Experimental design

I think that some of the experimental design is lacking.
For example to include the negative controls in the figures.
I do think that they need to show still Fc binding of both antibodies and if they eliminate the glycosilation of the original one, do they obtained the same results as the new one?
When something is Similar...it has to be similar to another thing. a lot of results are just described as "similar"...to what?
I do not agree in just using a predictive software for glycosilation, or lack of it. But if they would give more emphasis in showing that just this point mutation is sufficient for an antibody to loose its glycosilation, then I can accept it.
Regarding their DNA PCR product electrophoresis for IgG subtyping, how can you differentiate between a band of 211 bp for IgG1 and 207 bp for IgG3?
Some trivial Methods are described extensively, and most important methods are not. As a virologist, moi description is mandatory for a FRNT test.

Validity of the findings

The results are ok. Data is ok, conclusions can be more extensive and emphatic.
But their English level do not allowed the reviewer or the reader to understand everything thoroughly. One examplae of many: "To determine NT and ADE activity in Fc receptor bearing K562 cell, B3B9 rIgG was able to neutralize DENV2 and 3, but not DENV1 and 4".There is no sense in this sentence.

Additional comments

Please give the manuscript to an English native speaker for revision. In such a state at it is know, publication is not possible.

---

## Round 0.3 · Minor Revisions

I appreciate very much your time and changes performed to this version, which has improved over the originally submitted work. I encourage you to further modify your manuscript by following the reviewer´s comments and particularly in their annotated manuscript.

·

Basic reporting

The authors claim that the manuscript was proofed by a native speaker and by a professional service. The manuscript has improved, but there are still some mistakes and poor choices of words (I marked everything that needs to be changed in the enclosed document)
Results are still relevant and can be published.

Experimental design

There are some experimental details that need to be stated. Please take a look in the document attached.

Validity of the findings

The last sentence is a bit too out far reaching stating that the antibody produced by the authors can be used in therapy. I would say that it can be used in models testing for a possible anti dengue therapy.

Additional comments

Manuscript has improved, but needs still some polishing before publication.

---

## Round 0.4 · accepted · Accept

I would like to request that a native English speaker proofreads your manuscript before publication.

Reviewer 1 ·

Basic reporting

no comment

Experimental design

no comment

Validity of the findings

Lines 460-463, the authors should indicate that these are in vitro results and it is a proof of concept.
Additionally, the authors should be clear on potential "problem" of a human anti-mouse antibody response. The sentence should be reworded for clarity and good scientific presentation.
Lines 464-468, the authors should be clear on the conclusion. Try to include the phrase "with further studies on...." and emphasize on the potential utility of the antibody.
In the abstract, please remove the word "fortunately".

Additional comments

Generally the manuscript is in good order, the authors would need to do a thorough proof-reading, with a native English speaker. There are some minor grammar mistakes throughout the manuscript, eg line 396 (balancing vs balance); 472 (advisement vs advise).